# ChatCam: Empowering Camera Control through Conversational AI

**Xinhang Liu**[1]    **Yu-Wing Tai**[2]    **Chi-Keung Tang**[1]
[1]HKUST    [2]Dartmouth College

## Abstract

Cinematographers adeptly capture the essence of the world, crafting compelling visual narratives through intricate camera movements. Witnessing the strides made by large language models in perceiving and interacting with the 3D world, this study explores their capability to control cameras with human language guidance. We introduce *ChatCam*, a system that navigates camera movements through conversations with users, mimicking a professional cinematographer's workflow. To achieve this, we propose *CineGPT*, a GPT-based autoregressive model for text-conditioned camera trajectory generation. We also develop an *Anchor Determinator* to ensure precise camera trajectory placement. *ChatCam* understands user requests and employs our proposed tools to generate trajectories, which can be used to render high-quality video footage on radiance field representations. Our experiments, including comparisons to state-of-the-art approaches and user studies, demonstrate our approach's ability to interpret and execute complex instructions for camera operation, showing promising applications in real-world production settings. We will release the codebase upon paper acceptance.

## 1   Introduction

Cinematographers skillfully capture the essence of the 3D world by maneuvering their cameras, creating an array of compelling visual narratives [8]. Achieving aesthetically pleasing results requires not only a deep understanding of scene elements and their interplay but also meticulous execution of techniques.

Recent progress of large language models (LLMs) [1] has marked a significant milestone in AI development, demonstrating their capability to understand and act within the 3D world [29, 30, 87]. Witnessing this evolution, our work explores the feasibility of empowering camera control through conversational AI, thus enhancing the video production process across diverse domains such as documentary filmmaking, live event broadcasting, and virtual reality experiences.

Although the community has devoted considerable effort to controlling the trajectories of objects and cameras in video generation approaches for practical usage [4, 82, 75, 28], or predicting similar sequences through autoregressive decoding processes [35, 64], generating camera trajectories has yet to be explored. This task involves multiple elements such as language, images, 3D assets, and, beyond mere accuracy, necessitates visually pleasing rendered videos as the ultimate goal.

We propose *ChatCam*, a system that allows users to control camera operations through natural language interaction. As illustrated in Figure 1, leveraging an LLM agent to orchestrate camera operations, our method assists users in generating desired camera trajectories, which can be used to render videos on radiance field representations such as NeRF [52] or 3DGS [36].

At the core of our approach, we introduce *CineGPT*, a GPT-based autoregressive model that integrates language understanding with camera trajectory generation. We train this model using a paired text-trajectory dataset to equip it with the ability for text-conditioned trajectory generation. We also

38th Conference on Neural Information Processing Systems (NeurIPS 2024).

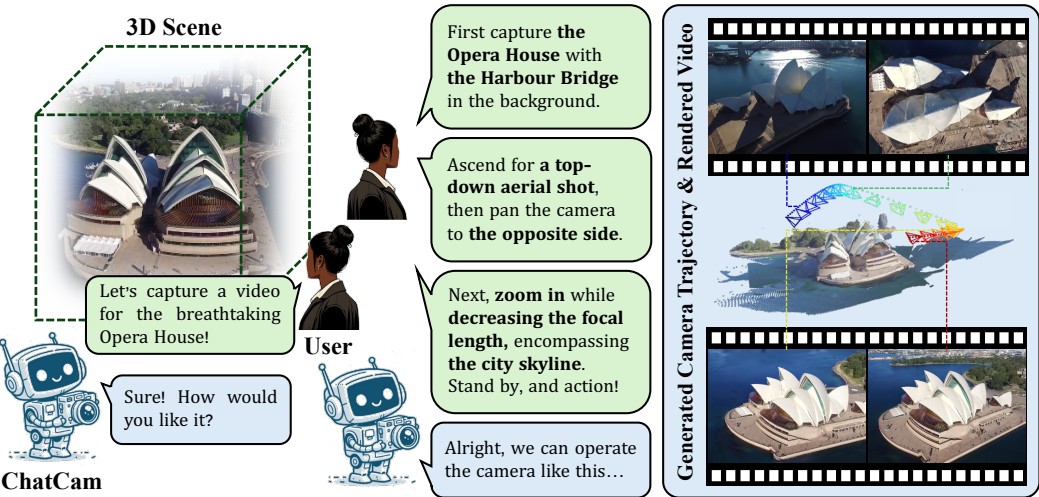

Figure 1: **Empowering camera control through conversational AI.** Our proposed *ChatCam* assists users in generating desired camera trajectories through natural language interactions. The generated trajectories can be used to render videos on radiance field representations such as NeRF [52] or 3DGS [36].

propose an *Anchor Determinator*, a module that identifies relevant objects within the 3D scene to serve as anchors, ensuring correct trajectory placement based on user specifications. Our LLM agent parses compositional natural language queries into semantic concepts. With these parsed sub-queries as inputs, the agent then calls our proposed *CineGPT* and *Anchor Determinator*. It composes the final trajectory with the outputs from these tools, which can ultimately be used to render a video that fulfills the user's request.

With comprehensive evaluations and comparisons to other state-of-the-art methods, our method exhibits a pronounced ability to interpret and execute complex instructions for camera operation. Our user studies further demonstrate its promising application prospects in actual production settings. In summary, this paper's contributions are as follows:

- We introduce *ChatCam*, a system that, for the first time, enables users to operate cameras through natural language interactions. It simplifies sophisticated camera movements and reduces technical hurdles for creators.

- We develop *CineGPT* for text-conditioned camera trajectory generation and an *Anchor Determinator* for precise camera trajectory placement. Our LLM agent understands users' requests and leverages our proposed tools to complete the task.

- Extensive experiments demonstrate the effectiveness of our method, showing how AI can effectively collaborate with humans on complex tasks involving multiple elements such as language, images, 3D assets, and camera trajectories.

## 2 Related Work

**Multimodal Language Models.** Large-scale language models (LLMs) [9, 19, 17, 1, 70] enabled by extensive datasets and model size, have demonstrated surprising emerging abilities. The emergence of multimodal models [43, 42, 33] is captivating as they can process text alongside other modalities such as images [25], audio [27], and videos [78]. Some unified models can perceive inputs and generate outputs in various combinations of text, images, videos, and audio [51, 68, 77, 86]. LLMs hold the potential to act as agents [76, 80, 62], allowing them to be driven by goals, reason about their objectives, devise plans, utilize tools, and interact with and gather feedback from the environment. Our proposed method involves multiple modalities including language, images, 3D fields, and camera trajectories, and utilizes LLMs as agents to assist users in operating cameras.

**Radiance Field Representations.** Utilizing continuous 3D fields modeled by MLPs and volumetric rendering, Neural Radiance Fields (NeRFs) [52] achieved breakthrough for novel view synthesis.

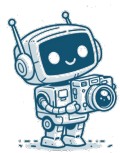
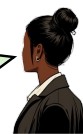

From outside the window, enter the bedroom and explore its layout following an S-shaped path.

**Observation:** The user is requesting a camera trajectory, starting from a location outside the window and then covering the bedroom's layout following an S-shaped path.

**Reasoning:** I need to find a starting point outside the window, then move through the window into the room and capture the room using an S-shaped path. Additionally, I should use a smooth panning speed to better showcase the room's layout.

**Plan:**

1. Call Anchor Determinator with *"capture the bedroom from outside the window"*.

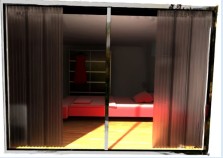

2. Call CineGPT with *"pan straight forward, smooth panning speed"*.

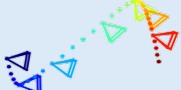

3. Call CineGPT with *"an S-shaped path, smooth panning speed"*.

4. Connect output trajectories from steps 2 and 3, ensuring they pass through the anchor point determined in step 1.

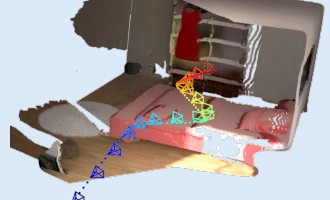

5. Ask the user for further clarification if needed.

6. Render a video using the generated camera trajectory.

Figure 2: **Overview of the ChatCam pipeline.** Given a camera operation instruction, ChatCam reasons the user's request and devises a plan to generate a trajectory using our proposed CineGPT and Anchor Determinator. The agent then utilizes the outputs from these tools to compose the complete trajectory and render a video.

Subsequent research has emerged to improve NeRFs and broaden their applications [69], including enhancing rendering quality [5, 6, 7**?** , 47], modeling dynamic scenes [84, 55, 58, 71, 74, 10, 22, 2**?** ], improving computational efficiency [83, 23], and facilitating 3D scene editing [45, 84, 73, 34, 46, 40]. Replacing the deep MLPs with a feature voxel grid network has demonstrated enhancement in both training and inference speed [13, 66, 24, 54]. More recently, 3D Gaussian Splatting [36] has further advanced visual quality and rendering efficiency. Compared to traditional 3D representations, radiance field representations offer superior photorealistic rendering quality, therefore, this study focuses on camera manipulation upon mainstream radiance field representations such as NeRF or 3DGS.

**3D Scene Understanding.** Early methods for 3D semantic understanding [32, 67, 79, 15] primarily focused on the closed-set segmentation of point clouds or voxels. NeRF's capability to integrate information from multiple viewpoints has spurred its application in 3D semantic segmentation [88, 20, 46, 53, 65, 26, 60, 31, 48, 49, 21]. Among these, [40, 37, 12] combine image embeddings from effective 2D image feature extractors [41, 11, 59, 39] to achieve language-guided object localization, segmentation, and editing. [21] proposes semantic anisotropic Gaussians to simultaneously estimate geometry, appearance, and semantics in a single feed-forward pass. Another line of research integrates 3D with language models for tasks such as 3D question answering [3], localization [14, 57, 81], and captioning [16]. Additionally, [29, 30, 87] propose 3D foundation models to handle various perception, reasoning, and action tasks in 3D environments. However, the AI-assisted operation of cameras within 3D scenes remains an unexplored area.

**Trajectory Control and Prediction.** Controlling the trajectories of objects and cameras is crucial to advance current video generation approaches for practical usage. TC4D [4] incorporates trajectory control for 4D scene generation with multiple dynamic objects. Direct-a-Video [82], MotionCtrl [75],

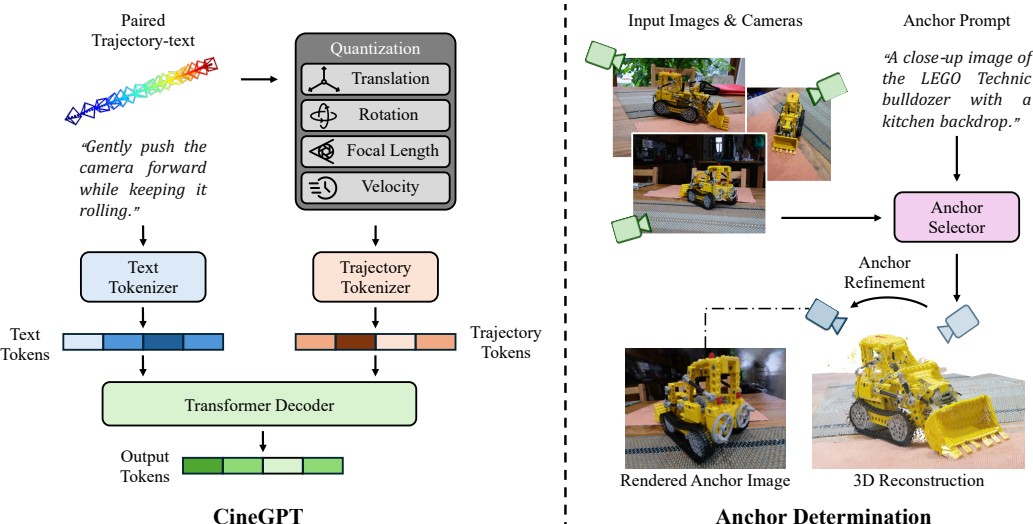

Figure 3: **(a) CineGPT.** We quantize camera trajectories to sequences of tokens and adopt a GPT-based architecture to generate the tokens autoregressively. Learning trajectory and language jointly, CineGPT is capable of text-conditioned trajectory generation. **(b) Anchor Determination.** Given a prompt describing the image rendered from an anchor point, the anchor selector chooses the best matching input image. An anchor refinement procedure further fine-tunes the anchor position.

and CameraCtrl [28] manage camera pose during video generation; however, they are either limited to basic types or necessitate fine-tuning of the video diffusion model. Moreover, these approaches require user-provided trajectories, whereas we, for the first time, generate camera trajectories conditioned on text.

## 3 Method

Figure 2 provides an overview of our method's pipeline. ChatCam analyzes the user's camera operation instruction and devises a plan to generate a trajectory using our proposed CineGPT and Anchor Determinator. Finally, an AI agent utilizes the outputs from these tools to compose the complete trajectory.

### 3.1 Text-Conditioned Trajectory Generation

To enable text-conditioned trajectory generation, we collect a text-trajectory dataset and introduce CineGPT, a GPT-based autoregressive model integrating language and camera trajectories. Illustrated in Figure 3 (a), our method quantizes camera trajectories into a sequence of trajectory tokens using a trajectory tokenizer. Subsequently, a multi-modal transformer decoder is employed to convert input tokens into output tokens. Upon training, our model adeptly generates token sequences based on user-provided text prompts. These sequences are then de-quantized to reconstruct the camera trajectory.

**Camera Trajectory Parameterization.** For each single frame, our camera parameters include rotation $\mathbf{R} \in \mathbb{R}^{3 \times 3}$, translation $\mathbf{t} \in \mathbb{R}^3$, and intrinsic parameters $\mathbf{K} \in \mathbb{R}^{3 \times 3}$. We further convert the rotation matrix $\mathbf{R}$ into the $\mathbb{S}^2 \times \mathbb{S}^2$ space [89] to facilitate computational efficiency and simplify the optimization process. The total $M$-frame camera trajectory is formulated as:

$$c_{1:M} = \{c_i\}_{i=1}^M = \{(\mathbf{R}_i, \mathbf{t}_i, \mathbf{K}_i)\}_{i=1}^M. \tag{1}$$

To additionally model the velocity of camera movement, we introduce a global parameter $t$ representing the total duration. Consequently, the instantaneous velocity of each frame can be approximated by the relative translation and rotation to the previnous frame over unit time.

**Text-Trajectory Dataset.** Given the scarcity of readily available data on camera operations, we manually constructed approximately 1000 camera trajectories using Blender [18]. These trajectories encompass a diverse range of movements, including various combinations of translations, rotations,

focal lengths, and velocities. Each trajectory is accompanied by a human language description detailing the corresponding movements. This dataset spans various scenarios, capturing both simple pan-tilt-zoom motions and more complex trajectories mimicking real-world scenarios.

**Trajectory Tokenizer.** We leverage a trajectory tokenizer based on the Vector Quantized Variational Autoencoders (VQ-VAE) architecture [72] to represent camera trajectories as discrete tokens. Our trajectory tokenizer consists of an encoder $\mathcal{E}$ and a decoder $\mathcal{D}$. Given an $M$-frame camera trajectory $c_{1:M} = \{c_i\}_{i=1}^M$, the encoder $\mathcal{E}$ encodes it into $L$ trajectory tokens $z_{1:L} = \{z_i\}_{i=1}^L$, where $L = M/l$ and $l$ is the temporal downsampling rate. The decoder $\mathcal{D}$ then decodes $z_{1:L}$ back into the trajectory $\hat{c}_{1:M} = \{\hat{c}_i\}_{i=1}^M$. Specifically, the encoder $\mathcal{E}$ first encodes frame-wise camera parameters $c_{1:M}$ into a latent vector $\hat{z}_{1:L} = \mathcal{E}(c_{1:M})$, by performing 1D convolutions along the time dimension. We then transform $\hat{z}_{1:L}$ into a collection of codebook entries $z$ through discrete quantization. The learnable codebook $Z = \{z_i\}_{i=1}^K$ consists of $K$ latent embedding vectors, each with dimension $d$. The quantization process $Q(\cdot)$ replaces each row vector with its nearest codebook entry, as follows:

$$z_i = Q(\hat{z}_i) = \arg \min_{z_k \in Z} ||\hat{z}_i - z_k||_2^2, \tag{2}$$

where $|| \cdot ||_2$ denotes the Euclidean distance. After quantization, the decoder projects $z_{1:L}$ back to the trajectory space as the reconstructed trajectory $\hat{c}_{1:M} = \mathcal{D}(z_{1:L})$. In addition to the reconstruction loss, we adopt embedding loss and commitment loss similar to those proposed in [85] to train our trajectory tokenizer. With a trained trajectory tokenizer, a camera trajectory $c_{1:M}$ can be mapped to a sequence of trajectory tokens $z_{1:L}$, facilitating the joint representation of camera trajectory and natural language for text-conditioned trajectory generation.

**Cross-Modal Transformer.** We utilize a cross-modal transformer decoder to generate output tokens from input tokens, which may consist of text tokens, trajectory tokens, or a combination of both. These output tokens are subsequently converted into the target space. To train our decoder-only transformer, we denote our source tokens as $X_s = \{x_s^i\}_{i=1}^{N_s}$ and target tokens as $X_t = \{x_t^i\}_{i=1}^{N_t}$. We feed source tokens into it to predict the probability distribution of the next potential token at each step $p_\theta(x_t|x_s) = \prod_i p_\theta(x_t^i|x_t^{<i}, x_s)$. The objective function is formulated as:

$$\mathcal{L}_{\text{LM}} = -\sum_{i=1}^{N_t} \log p_\theta(x_t^i|x_t^{<i}, x_s). \tag{3}$$

By optimizing this objective, we aim to equip CineGPT with the ability to capture intricate patterns and relationships within the data distribution. We then fine-tune CineGPT on supervised trajectory-language translation leveraging our paired text-trajectory dataset, where the input for this stage can either be a camera trajectory or a text description, while the target is the opposite modality. During inference, CineGPT can generate camera trajectories solely from textual descriptions as inputs.

## 3.2 Object-Centric Trajectory Placement with Anchors

While CineGPT enables text-conditioned trajectory generation, its generation process solely focuses on determining the camera's movements, without contextual connection to specific scenes. Consequently, CineGPT alone cannot effectively handle user prompts that involve object-centric descriptions, such as directives like "directly above the Sydney Opera House". In this light, we bridge trajectory generation with each underlying scene with "anchors" serving as reference points within the scene to achieve more accurate placement of trajectories, as illustrated in Figure 3 (b).

Our anchor determination procedure takes natural language descriptions of an image as input. This procedure identifies a set of camera parameters that can render an image that best matches the given description. Current 3D visual grounding approaches [57, 81] typically entail learning a 3D feature field [40, 37] and localizing objects within the scene, which often results in high computational costs. In contrast, our anchor determinator adopts a different strategy. Initially, it selects the input image that best matches the given text description as an initial anchor. Subsequently, an anchor refinement process is employed to iteratively improve upon this initial anchor, ultimately yielding the final anchor. This approach offers a more efficient alternative to traditional methods, reducing computational overhead while still achieving accurate scene anchoring.

**Initial Anchor Selector.** Since our method leverages radiance field representations to render videos, we naturally have access to the input images for training the 3D scene representations. We utilize

an initial anchor selector based on CLIP [59] to choose the image from these input images that best matches the text prompt. To be specific, for $i$-th input image $I_i$, we extract their CLIP image features and convert the text prompt $T$ into a CLIP text feature. Next, we compute the cosine similarity between the CLIP text feature vector and each of the CLIP image feature vectors. We select the best matching image with the highest cosine similarity score as the initial anchor. This can be formulated as:

$$i_{\text{anchor}} = \arg\max_i \frac{f_{\text{image}}(I_i) \cdot f_{\text{text}}(T)}{\|f_{\text{image}}(I_i)\|\|f_{\text{text}}(T)\|}, \tag{4}$$

where $f_{\text{image}}(\cdot)$ and $f_{\text{text}}(\cdot)$ represent the image and text feature extractor, respectively.

**Anchor Refinement.** Using the camera parameters $c_{\text{anchor}}$ associated with the selected image as initialization, we further minimize the following objective to obtain the final anchor camera parameters:

$$\min_c \mathcal{L}_{\text{anchor}}(c) = -\frac{f_{\text{image}}(R(c)) \cdot f_{\text{text}}(T)}{\|f_{\text{image}}(R(c))\|\|f_{\text{text}}(T)\|}, \tag{5}$$

where $R(\cdot)$ is the rendering function and $c$ is initialized with $c_{\text{anchor}}$. The optimization of $c$ is performed using gradient descent, with the update rule given by:

$$c_{t+1} = c_t - \eta \nabla_c \mathcal{L}_{\text{anchor}}(c_t), \tag{6}$$

where $\eta$ is the learning rate. The optimization typically achieves convergence within 100 to 1000 steps. This refinement process ensures that the camera parameters are adjusted to better match the text prompts, handling cases where the initial input images do not align well with the prompts.

### 3.3   Trajectory Generation through User-Friendly Interaction

With our proposed CineGPT and anchor determination, a large language model acts as an agent to interpret the user's requests, generates a plan to use various tools, and composes a final camera trajectory. We adopt GPT-4 [1] to interpret users' natural language inputs and subsequently produce trajectory prompts. Specifically, we use a carefully designed prompt to instruct the LLM agent to reason about the user's requirements and devise a plan consisting of the following steps: 1) Break down the complex text query into sub-tasks that CineGPT and the Anchor Determinator can effectively handle. 2) Use these tools to generate atomic trajectories and determine anchor points. 3) Compose the final trajectory by concatenating atomic trajectories and ensuring they pass through the anchors.

**Observing, Reasoning, and Planning.** Research indicates that LLMs can be prompted to decompose complex goals into sub-tasks, essentially thinking step-by-step [76]. As illustrated in Figure 2, we begin by instructing the agent to describe its observations, providing a summary of the current situation. The agent then uses this summary to reason and develop a mental scratchpad for high-level planning. Finally, it outlines specific steps to achieve the overarching goal of generating the user-required camera trajectory.

**Utilization of Proposed Tools.** We inform our agent of the expected input and output format, i.e., the APIs, of our proposed CineGPT and Anchor Determinator, and instruct the agent to interact with them following the given format. In its outlined specific steps to generate the user-required camera trajectory, it first calls CineGPT and Anchor Determinator to obtain atomic trajectories and anchor points, respectively. Note that both tools can be called multiple times, and multiple atomic trajectories can later be concatenated into final trajectories that pass through all anchor points correctly.

**Final Trajectory Composition.** Here we explain how to combine atomic trajectories from CineGPT with anchor points to form the final trajectory. The agent first decides the role of the anchors in the ultimate trajectory, either as a starting point or an ending point of some atomic trajectory. Then affine transformations are applied to the respective atomic trajectories to ensure that their starting or ending points align with the anchor points. For the remaining atomic trajectories not controlled by anchor points, affine transformations are applied to make the endpoint of the previous trajectory align with the starting point of the subsequent trajectory.

## 4   Experiments

We assess the performance of our proposed ChatCam for human language-guided camera operation across a series of challenging scenarios. Through ablation studies, we provide empirical evidence of

*Zoom in from directly above the Sydney Opera House and roll the camera.*

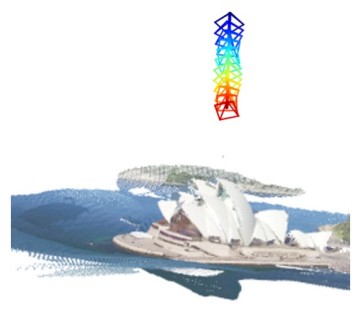

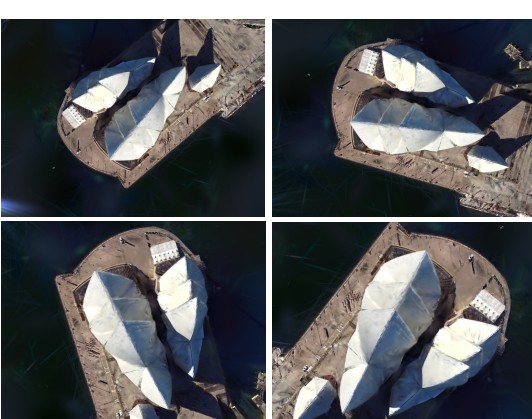

*Do a dolly zoom focusing on the chandelier.*

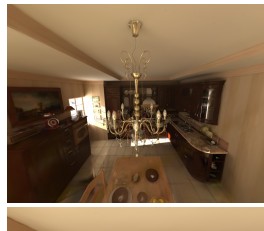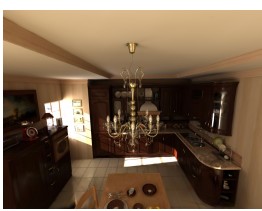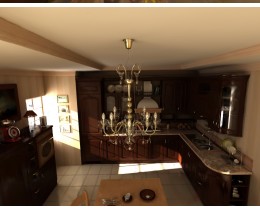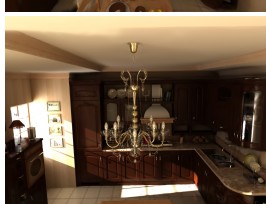

*Pan the camera from left to right along the piano, then turn to look at the bicycle on the right.*

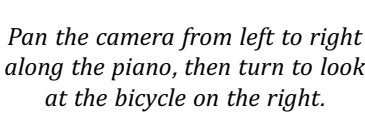

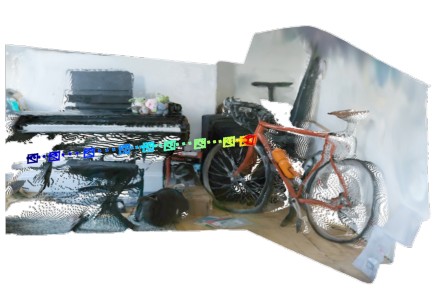

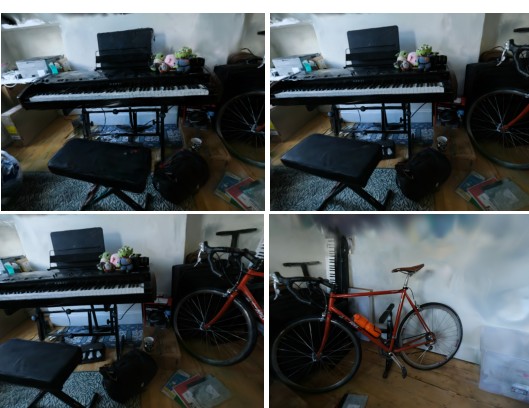

Figure 4: **Qualitative results on indoor and outdoor scenes.** Visualizations of our generated trajectories from input text descriptions and the frames in the final rendered video. Our method is capable of understanding and executing instructions and providing correct translations, rotations, and camera focal lengths. Additionally, our method can comprehend more specialized terms such as "dolly zoom".

the effectiveness of its fundamental components. We kindly refer the reader to our supplementary material for additional experimental results, including rendered **videos**.

## 4.1 Experimental Setup

**Implementation Details.** We implement our approach using PyTorch [56] and conduct all the training and inference on a single NVIDIA RTX 4090 GPU with 24 GB RAM. The trajectory tokenizer has a codebook with $K = 256$ latent embedding vectors, each with dimension $d = 256$. The temporal downsampling rate of the trajectory encoder is $l = 4$. Our cross-modal transformer decoder consists of 24 layers, with attention mechanisms employing an inner dimensionality of 64.

*Sweep across the boy in black, the keyboard, and the boy in white, then zoom out to frame the boys and the white guitar together.*

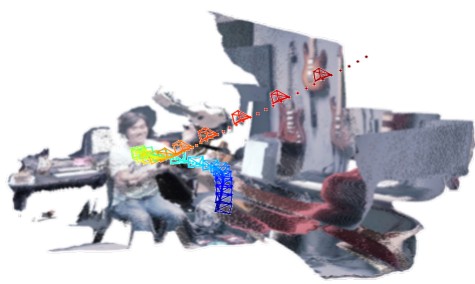

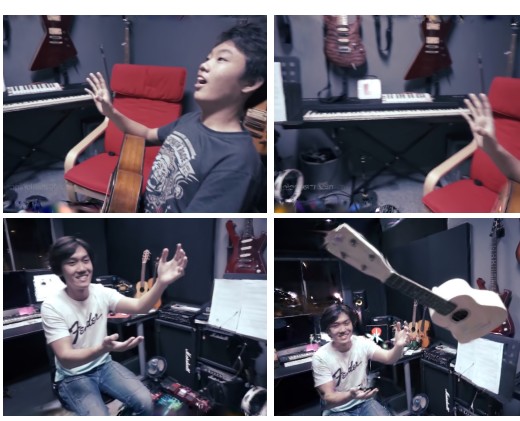

*Starting from the man playing cards, pan the camera along the corridor, then back up to the end and look to the right.*

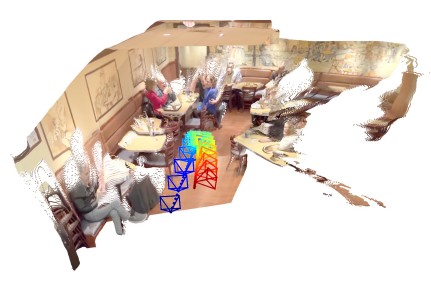

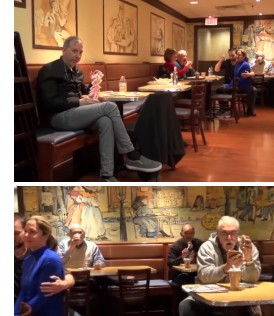

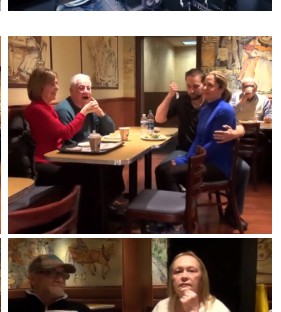

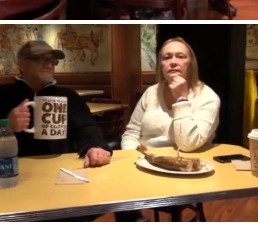

Figure 5: **Qualitative results on human-centric scenes.** Visualizations of our generated trajectories from input text descriptions and the frames in the final rendered video. Our method performs effectively in scenes with multiple humans.

The remaining sub-layers and embeddings have a dimensionality of 256. We train CineGPT using the Adam optimizer [38] with an initial learning rate of 0.0001. It takes approximately 30 hours to converge. Our anchor determination utilizes CLIP [59] with a ViT-B/32 Transformer architecture. The learning rate of anchor refinement is 0.002. By default, we use GPT-4 [1] as our LLM agent, and its prompt will be released with our codebase. We render final videos using 3DGS [36] as the 3D representation.

**Tested Scenes.** We tested our method on scenes from a series of datasets suitable for 3D reconstruction with radiance field representations, including: (i) *mip-NeRF 360* [6], a real dataset with indoor and outdoor scenes. (ii) *OMMO* [50], a real dataset with large-scale outdoor scenes. (iii) *Hypersim* [61], a synthetic dataset for indoor scenes. (iv) *MannequinChallenge* [44], a real dataset for human-centric scenes. If camera poses associated with images were not provided, we used COLMAP [63] for camera pose estimation. For each scene, we reconstructed using all available images without train-test splitting.

**Baselines.** As the first method to enable human language-guided camera operation, there is no established direct baseline for comparison. Therefore, we adopt 3D understanding approaches based on radiance field representations to let the LLM agent attempt to select a series of images corresponding to the input text from input images and interpolate their camera poses to construct camera trajectories. These methods include LERF [37], utilizing CLIP embeddings, and SA3D [12], utilizing SAM embeddings.

**Evaluation Metrics.** To evaluate the accuracy of the generated trajectories, we manually construct ground truth trajectories and compute the mean squared errors (MSEs) of translations and rotations relative to them. Additionally, we conduct a user study to evaluate the rendered videos using generated camera trajectories, where users are asked to select the video with the best **visual quality** and best **alignment** with the input text.

*Facing the piano, pull the camera back, then glance over at the TV on the left, and back to the piano.*

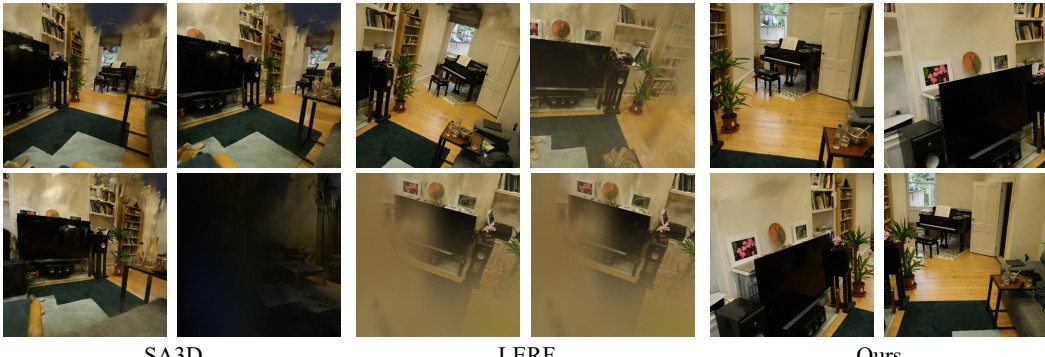

| SA3D | LERF | Ours |

Figure 6: **Qualitative comparisons.** Our approach avoids moving the camera to unreasonable positions such as inside objects, obtaining videos with better visual effects, and aligning best with input texts.

Table 1: **Quantitative comparisons and evaluations.** Our full model performs better than baselines and variants in terms of trajectory accuracy, visual quality, and alignment with input text.

| Method | LLM Agent | Anchor Determination | Translation MSE ($\downarrow$) | Rotation MSE ($\downarrow$) | Visual Quality ($\uparrow$) | Alignment ($\uparrow$) |
|---|---|---|---|---|---|---|
| SA3D [12] | *GPT-4* | - | 19.5 | 6.3 | 5.7 | 3.8 |
| LERF [37] | *GPT-4* | - | 17.7 | 4.9 | 9.4 | 28.3 |
| ChatCam (Ours) | *LLaMA-2* | ✓ | 6.4 | 3.6 | - | - |
| ChatCam (Ours) | *GPT-3.5* | ✓ | 7.3 | 3.5 | - | - |
| ChatCam (Ours) | *GPT-4* | ✗ | 16.2 | 8.5 | - | - |
| **ChatCam (Ours)** | *GPT-4* | ✓ | **5.3** | **2.9** | **84.9** | **67.9** |

## 4.2 Results

As shown in Figure 4, our method demonstrates the ability to understand and execute camera operation instructions on a range of complex indoor and outdoor scenes, giving appropriate translation, rotation, and focal length. Our method also understands more technical terms such as dolly zoom, which creates a special visual effect by zooming the camera out while adjusting the focus. In Figure 5 we further showcase the qualitative results of our method in human-centric scenes. Our method can correctly handle user instructions about specific people and create correct and vivid visual effects.

**Comparisons.** In Figure 6 we qualitatively compare our method with LLM agents utilizing SA3D or LERF to locate target objects. The baselines do simple interpolation of keyframes because they have no knowledge about camera trajectories and tend to move the camera to unreasonable spots (such as entering an object). Therefore, the video rendered by baselines contains artifacts and is not correctly consistent with the input text. However, our method achieves better visual quality and alignment with input texts. Quantitative comparisons in Table 1 further prove that our method has better performance and is preferred by users.

**Ablation Study.** We present our ablation study in Table 1. We evaluate the performance of our method using different LLMs as agents. Our approach achieved the best accuracy using GPT-4 [1] as the agent, better than GPT-3 [9] and LLaMA-2 [70].Without our proposed anchor determination, our method cannot correctly place trajectories within 3D scenes, thereby being less accurate than our full model.

## 5 Conclusion

This paper presents ChatCam, a system designed for camera operation through natural language interactions. By introducing CineGPT, we bridge the gap between human language guidance and camera control, achieving text-conditioned trajectory generation. Our proposed anchor determination procedure further ensures precise camera trajectory placement. Our LLM agent comprehends users' requests and effectively utilizes our proposed tools to compose the final trajectory. Through extensive experiments, we demonstrate the effectiveness of ChatCam, showcasing its ability to collaborate with humans on complex tasks involving language, images, 3D assets, and camera trajectories. ChatCam has the potential to simplify camera movements and reduce technical barriers for creators.

## Acknowledgements

This work was supported in part by Dartmouth College A&S Startup fund and by the Research Grant Council of the Hong Kong SAR under Theme-based Research Scheme, grant no. T22-606/23R.

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

# Supplementary Materials for
## *ChatCam: Empowering Camera Control through Conversational AI*

This supplementary document presents additional qualitative results and discusses the limitations and societal impacts of our proposed approach.

## A    Video

For better visualization of our reconstruction results, we create a set of video visualizations. We highly recommend to watch *supplementary_video.mp4* for more results.

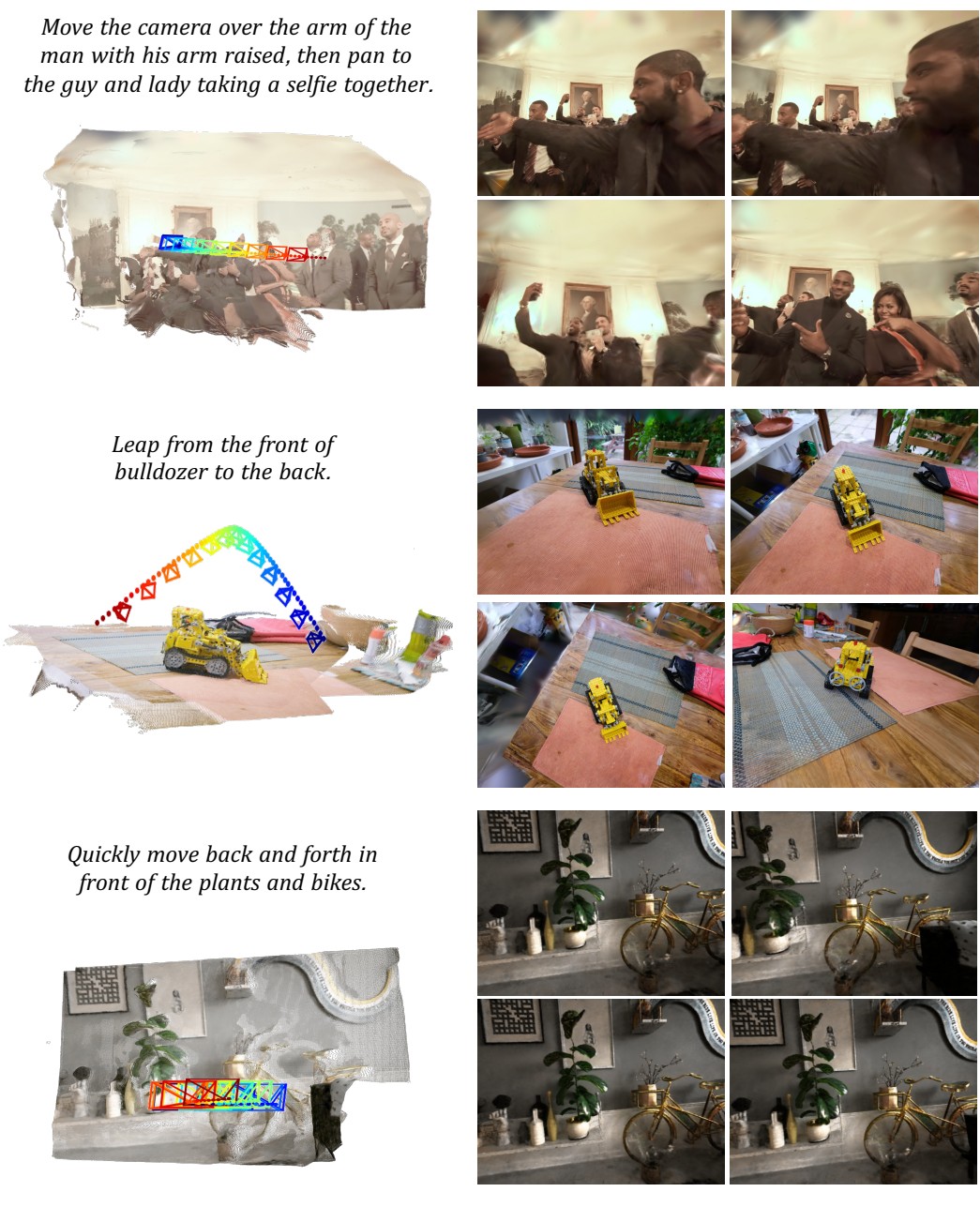

*Move the camera over the arm of the man with his arm raised, then pan to the guy and lady taking a selfie together.*

*Leap from the front of bulldozer to the back.*

*Quickly move back and forth in front of the plants and bikes.*

Figure A: **Additional qualitative results. (1)**

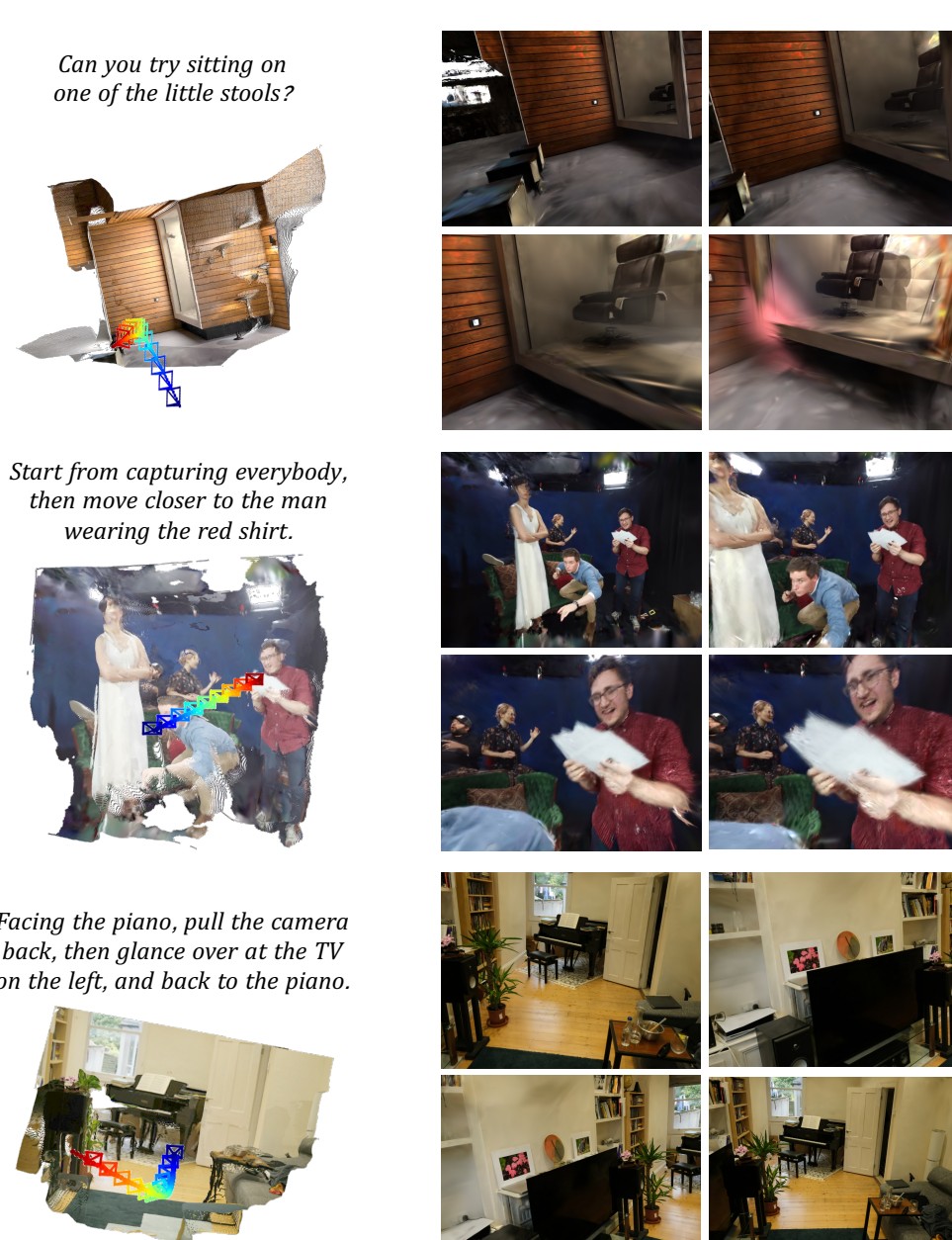

*Can you try sitting on one of the little stools?*

*Start from capturing everybody, then move closer to the man wearing the red shirt.*

*Facing the piano, pull the camera back, then glance over at the TV on the left, and back to the piano.*

Figure B: **Additional qualitative results. (2)**

## B  Additional Results

We present additional qualitative results in Figure A and Figure B.

## C  Limitations

As the first AI-assisted system for language-guided camera operation, our method relies on LLMs as agents, and therefore its efficiency depends on LLMs. With the rapid development of the community, this limitation may be alleviated.

Our current results are limited to static scenes due to the limited availability of high-quality 4D Dynamic NeRF/3DGS data. Extending our approach to a dynamic scene would be straightforward by introducing a timestamp in the Anchor Determinator. We leave this as one of our future work.

## D   Societal Impacts

Our approach has great potential to help creators in industries such as television, movies, games, etc. reduce their burden by simplifying the learning costs of utilizing 3D assets like radiance field reconstructions. This allows content creators to focus on their creations. We must also admit that as existing 3D assets become more and more abundant, it is inevitable that there will be harmful content in them, and our method may contribute to the creation of harmful content. We encourage the community to play wisely with ChatCam.

