# OpenReview forum: "ChatCam: Empowering Camera Control through Conversational AI"
_NeurIPS.cc/2024/Conference — NeurIPS 2024 poster_

### Official Review · Reviewer_oNWk · 2024-07-12

**Soundness:** 3
**Presentation:** 3
**Contribution:** 3
**Rating:** 6
**Confidence:** 4

**Summary:**

This paper introduces ChatCam, a system that enables camera operation via natural language interactions. This system has two key components. CineGPT is proposed for text-conditioned trajectory generation and an Anchor Determinator for precise camera trajectory placement. Experimental results illustrate ChatCam’s effectiveness in text-conditioned trajectory generation and show the potential to simplify camera movements and lower technical barriers for creators.

**Strengths:**

1.	ChatCam is a novel system to generate camera trajectory via natural language.
2.	The paper is well-written and easy to follow.
3.	The proposed GineGPT and Anchor Determinator are technically solid.

**Weaknesses:**

The system is only verified on several limited 3D static scenes, and the generated trajectory is relatively short.

Some components are not fully verified. For example, what’s the effect of the text prompt “smooth panning speed” in Fig. 3? How does this text prompt affect the trajectory?

Since the method was only evaluated on a small dataset, and the trajectory seems short, it's not clear whether the generations overfit to a small set of text prompts.

Some experimental details are unclear.
1.	How about the training dataset? L114 suggests there are 1000 trajectories to train the trajectory generation. I would suggest introducing the training dataset in detail in Sec 4.
2.	What’s the time cost of Anchor Determination in inference?

**Questions:**

See limitations.

1. Reference information inaccurate: [32,34], NeurIPS 2023 paper, not NeurIPS 2024

2. For the same text prompt, ChatCam is expected to generate different trajectories for different scenes. A verification is interesting to illustrate the effectiveness of an Anchor Determinator.

3. How long is the trajectory sequence?

4. It’s not clear whether the approach works for dynamic scenes, but it seems the Anchor Determinator does not model the temporal relations.

**Limitations:**

Yes

---

> ### Author Rebuttal · Authors · 2024-08-07
>
> Thank you for your positive feedback as well as thoughtful suggestions and questions. Below, we address your points individually.
>
> **Limited Scenes & Overfitting & Trajectory Length.** The cases we test cover "a wide range of trajectories on complex scenes (indoor/outdoor, object/human-centric) with several interesting elements", as attested by reviewer **dvT5**. We do not observe significant overfitting, due to the comprehensive dataset and the leverage of CLIP. Furthermore, ChatCam allows users to correct suboptimal results through conversations, further mitigating possible errors.
>
> The length of each trajectory in our dataset is about 60 frames, matching the length of a single inference of CineGPT. ChatCam has no limit on the length of the trajectory, and users can continue to add requirements to infer CineGPT multiple times and extend trajectories.
>
> Please also refer to **the new example we provide in the general response**, where the trajectory involves panning downstairs, passing through a tunnel, performing a U-turn, and executing a dolly zoom within a large complex room.
>
> **Prompts about Speed.** Prompts such as "smooth panning speed" affect the speed of the trajectory generated by CineGPT. Specifically, for the same duration (frame number), trajectories can cover different distances. The table below illustrates the effect of such prompts.
> | Prompt               | Duration (frames) | Distance Covered |
> |----------------------|--------------------|------------------|
> | “Pan forward, slowly.”         | 60                 | 24.11       |
> | “Pan forward.”                 | 60                 | 64.5       |
> | “Pan forward, rapidly.”        | 60                 | 91.5       |
>
>
>
>
> **Validating Anchor Determinator.** In Figure D of the attached PDF, we qualitatively show how the same text prompt results in different trajectories in different scenes due to Anchor Determination.  Also, in the original submission, we conduct an ablation study on the Anchor Determinator, with quantitative results in Table 1.
>
>
>
> **Anchor Determinator Inference Time.** On a single NVIDIA RTX 4090 GPU with 24 GB RAM, a single inference of the Anchor Determinator takes approximately 10-15 seconds. The majority of this time is spent on anchor refinement rather than CLIP-based initial anchor selection.
>
> **Trajectory Dataset.** We will introduce the dataset in detail in Section 4.2 as suggested. We enumerate the textual descriptions of camera trajectories and then build a camera trajectory in Blender for each description. The camera movements included are basic translations, rotations, focal length changes, and combinations of these (simultaneous or sequential). We also include some camera trajectories mentioned in professional cinematography literature and build them in Blender. All camera trajectories cover similar volumes, with aligned translations and rotations in the first frame.
>
> **Dynamic Scenes.** CineGPT can be directly applied to dynamic scenes due to its independence from specific scenes. Extending our approach to dynamic scenes would be straightforward by introducing a timestamp in the Anchor Determinator so that the "anchor" contains temporal coordinates in addition to spatial coordinates. We will show results on dynamic scenes in an updated version.
>
> **Inaccurate References.** Thank you for pointing this out; we will correct them.

---

> > ### Comment · Reviewer_oNWk · 2024-08-13
> >
> > Thanks for the response. Most of my concerns have been solved.

---

> > > ### Author Response · Authors · 2024-08-13
> > > **Thank you for your response**
> > >
> > > Thank you for the response and we will revise the submission according to the reviewers' concerns.

---

### Official Review · Reviewer_Shd5 · 2024-07-13

**Soundness:** 3
**Presentation:** 3
**Contribution:** 3
**Rating:** 5
**Confidence:** 4

**Summary:**

This paper propose ChatCam, a pipeline that utilizes GPT to translate the natural language into professional camera trajectory, which enhances the video production process for common users.

**Strengths:**

1. This paper divide the task into three steps: observation, reasoning and planning. The first two steps is completed by GPT and get the instruction of every subtasks. In the third step, the author proposes Anchor Determinator and CineGPT to generate the initialization point and camera trajectory.
2. The qualitative and qualitative results and video demo proves the efficiency of this method.
3. The paper is well written and easy to follow.

**Weaknesses:**

The design of this paper is clever and makes good use of the advantages of LLM. But I still has some concerns. If the authors can address my concerns, I would change my rate.
1. In the planning part of Figure 2, how can you ensure there is no collision with 3D scenes in every sub-step, such as “an S-shaped path, smooth panning speed”. If the  structure of the room is complicated, it is easy to collide with the scenes.
2. Is natural language really the best way for users to interact？Maybe user can import the 3D GS of scenes into Unreal Engine, where Luma AI provide this plugin (https://www.unrealengine.com/marketplace/en-US/product/luma-ai), and generate the camera trajectory by adding camera key points in the 3D space by UI interface, with position, rotation, camera intrinsics. The frames between key points can be obtained by using bilinear interpolation. You can preview whether collision has occurred in real time, and you can also adjust camera parameters by drag. (https://dev.epicgames.com/documentation/en-us/unreal-engine/creating-camera-cuts-using-sequencer-in-unreal-engine)
3. Can you show more complicated camera trajectories, such as dolly. The “straight forward and roll”, “S-shaped path”, “from left to right”,  which just need coarse control, seems can be easily obtained by the item 2 I propose above (UE).

**Questions:**

If the generated trajectory is not satisfactory or contains collision, how can user interact with the ChatCam? Whether the pipeline can modify the generated trajectory?

**Limitations:**

The collision in the 3D scenes is not explicitly and hardly constrained in this pipeline.

---

> ### Author Rebuttal · Authors · 2024-08-07
>
> Thank you for your time and valuable feedback. While we are encouraged by your appreciation for our ideas and results, the issues you point out are crucial.
>
> **Collision.** Collisions can indeed occur, especially when the scene is complex or the user does not provide enough anchor points. To keep the pipeline simple yet effective, we do not handle collisions specifically. In particularly complex scenes, such as indoor scenes with multiple rooms, users can provide more detailed text descriptions to guide ChatCam in finding anchor points to avoid collisions. Moreover, if ChatCam generates trajectories that include collisions, users can guide it to make corrections through further dialogue, as discussed below and demonstrated in the example we provide in the general response, where ChatCam first generates a trajectory that collides with the wall, so the user avoids it by providing more anchor points.
>
> **User Interaction & Modification.** The user and ChatCam can engage in multiple rounds of dialogue to iterate and improve the generated trajectories, similar to interactions between a director and a photographer. We verify this advantage in the **example provided in the general response**. In this example, ChatCam first generates a trajectory that collides with the wall, so the user avoids the collision by providing more anchor points. Furthermore, the user proposes to extend the trajectory with a dolly zoom and
>  complete this improvement simply through dialogue.
>
> **Is Natural Language Really the Best Way for Users to Interact?** As discussed in the general response, ChatCam allows lay users without professional knowledge such as 3D rotation, translation, camera intrinsics, and keypoints to produce production-quality cinematography. ChatCam also benefits from a pre-trained LLM with multi-lingual understanding, enabling users to interact with ChatCam in languages other than English. Compared with professional software and plug-ins with UI interfaces, this greatly lowers the entry threshold for lay users.
>
> Through ChatCam, users can engage in further conversation with ChatCam to interactively and iteratively refine the target camera trajectory, much like how a human director gives natural language suggestions to a human cinematographer in a movie post-production process.
>
> Putting ChatCam into perspective, its contributions are not limited to current camera applications. We believe our explorations in conversational AI’s capability for multimodal reasoning and planning have important implications for building embodied AI agents.
>
> **More Complicated Trajectories.** In in the general response, we present **an additional example of a complex camera trajectory**. This trajectory involves panning downstairs, passing through a tunnel, performing a U-turn, and executing a dolly zoom within a complex room. We believe such intricate operations are not easily achievable by lay users through professional software and plug-ins with UI interfaces, and demonstrate "a wide range of trajectories on complex scenes with several interesting elements," as attested by reviewer **dvT5**.

---

> > ### Comment · Reviewer_Shd5 · 2024-08-13
> >
> > After reading other reviewers' comments and authors' response, most of my concerns are addressed. Conversation Example shows how to handle the collision by language in an iterative way, which is user-friendly. In the response to kHgE, they prove the efficiency of CineGPT compared to the interpolation and rule-based methods, which means this method requires fewer anchor points than existing tools. I hope authors could add these experiments and discussions in the final version. I would change my rate to borderline accept.

---

> > > ### Author Response · Authors · 2024-08-13
> > > **Thanks for your response**
> > >
> > > Thank you for your response! We greatly appreciate the valuable discussion and will revise our work based on the reviewers' suggestions.

---

### Official Review · Reviewer_dvT5 · 2024-07-15

**Soundness:** 3
**Presentation:** 3
**Contribution:** 4
**Rating:** 6
**Confidence:** 4

**Summary:**

The paper proposes a method for generating camera trajectories for rendering a 3D scene, conditioned on natural language. This problem statement is novel and original, very useful, and the paper demonstrates convincing results. The method uses two components: a language-conditioned camera trajectory generator, and a language-conditioned camera anchor generator. An LLM takes a high-level language query as input and creates a plan based on these two components.

**Strengths:**

The problem statement is novel and seems to be very useful with high potential impact.
The method is creative and novel.
Results are excellent. They demonstrate wide range of trajectories on complex scenes with several interesting elements.

**Weaknesses:**

The paper fails to demonstrate the planning output of the LLM. Is Fig. 2 an actual output of the method, or just an illustration? I would expect to see many more results of the actual plan used to generate the trajectories. It is very difficult to understand the workings of the method without these.

The method seems far from reproducible. Sec 3.3 talks about several important components of the method, but only at a very high-level that would not enable anyone to reproduce the results. The details of the finetuning proposed in L142-146 are also unclear.

The quantitative evaluations rely on ground truth trajectories. Shouldn't there be many possible GT trajectories consistent with a language query? How would this diversity impact the evaluations? The same holds true for output of the system. Can the method generate diverse outputs? If so, the paper should include such results, and ideally, should also update the metrics to reflect this.

In the output in Fig. 2, there does not seem to be an end anchor point. How would the method know the extent or volume of space the S-shaped path should cover? Are there many cases where the volume of the space is undetermined based on the plan?

Comparisons to baselines should also be included as videos, especially as the paper talks about their trajectories.

**Questions:**

What are the limitations of the approach? Does the LLM fail to produce reasonable plans at times? When does that happen? Is the scale of the trajectories always fully determined? Can you comment more on the failure cases?

**Limitations:**

Limitations are not mentioned

---

> ### Author Rebuttal · Authors · 2024-08-07
>
> Thank you for your time and valuable feedback. While we are encouraged by your appreciation of our efforts, the issues you point out are constructive. Below, we address them:
>
> **Conversation Example & Planning Output.** Figure 2 shows an actual conversation with the LLM (with visualizations and translation from JSON to human language by the authors). As suggested, in the **new example in the general response**, we show dialogue where users and ChatCam gradually correct and improve the output trajectory. At the same time, we also post our designed LLM prompt in the general response to help better understand such conversations.
>
> **Reproducibility.** To increase the reproducibility of trajectory generation through user-friendly interaction in Section 3.3, we post our designed **LLM prompt** in the general response and will make it public upon acceptance.
>
> L142-L146 describes the training of CineGPT. To provide more details:
> i) In the first stage, we perform unsupervised pre-training in an autoregressive manner, where it conditions on previous tokens in a sequence to predict the next token [1, 2]. This process involves maximizing the likelihood of the token sequences (Equation 3), allowing the model to capture relationships between tokens and generate coherent outputs.
> ii) In the second stage, we use the paired text-trajectory dataset to supervise CineGPT for text-to-trajectory and trajectory-to-text translation, to obtain the model weights we finally use.
>
> We are willing to answer any further technical questions to increase the reproducibility of our work.
>
> **Multiple Outputs/Ground Truths.** To strengthen our evaluation by considering multiple possible ground truth trajectories, we added a set of experiments where we built multiple GT trajectories. We calculated the distance between the predicted trajectory and the nearest GT trajectory using a new metric $\min_i \|\| x_{\text{pred}} - x_{\text{gt}_i}\|\|_2$ that we call Minimum Distance to Ground Truth (MDGT), and compare it with baselines. From the results in the table, it is evident that considering multiple possible GTs, our method still achieves the best quantitative results.
>
> Our method can generate diverse results. Therefore, we report the Frechet Inception Distance (FID), which measures the distribution discrepancy between the ground truth and generated trajectories, and Diversity (DIV), which evaluates the diversity of generated motion by calculating the variance from the extracted features from the trajectory encoder. We do not report FID and DIV for our baselines because they can only generate deterministic trajectories using simple interpolation.
>
> | Method          | Translation MDGT (↓)  | Rotation MDGT (↓)  | FID (↓)   | DIV (↑)   |
> |-----------------|-----------------------|--------------------|-----------|-----------|
> | SA3D            | 15.7                  | 4.7                | –         | –         |
> | LERF            | 13.3                  | 4.3                | –         | –         |
> | Ours            | 4.7                   | 2.1                | 0.89      | 7.0       |
>
> **Extent of Camera Movement.** The extent of paths like S-shaped ones depends heavily on CineGPT’s training data. Users can control the scale of the trajectory by using textual cues describing the scale. Additionally, users can provide explicit cues to add more anchors to adjust the scale, as demonstrated in the conversation example provided in our general response.
>
> **Baseline Result Video.** We have highlighted the artifacts and inaccuracies caused by the baseline output in the form of images. Due to the limitations of external links, we are unable to include new videos at this stage but will update the baseline result video in our future version.
>
> **Limitations and Failure Cases.** We have discussed the limitations of our method in the appendix, including its efficiency depending on LLMs and the lack of exploration of dynamic scenes. We will incorporate the limitations pointed out by other reviewers in the updated version. The most common failure case of our method occurs when the generated trajectory bumps into or goes through objects like walls or doors. Additionally, if the text prompt input to CineGPT is too complex, especially containing rare descriptions of shapes, it may not generate the correct trajectory. However, we kindly point out that these failure cases can be corrected by chatting with the LLM agent.
>
> **References:**
> 1. Radford et al., "Improving Language Understanding by Generative Pre-Training".
> 2. Radford et al., "Language Models are Unsupervised Multitask Learners".

---

> > ### Comment · Reviewer_dvT5 · 2024-08-13
> >
> > Thanks for the response. It addresses my concerns. The revised paper should clearly mention the issues with defining the scale of the trajectories and no collision handling, and show corresponding results.

---

> > > ### Author Response · Authors · 2024-08-13
> > > **Thanks for your response**
> > >
> > > Thank you for your response and for appreciating our research! We will mention these issues and include the experimental results in the revised paper. We will also incorporate the reviewers' other suggestions to further strengthen our work.

---

### Official Review · Reviewer_kHgE · 2024-07-16

**Soundness:** 3
**Presentation:** 3
**Contribution:** 3
**Rating:** 4
**Confidence:** 5

**Summary:**

The paper proposes a method to generate camera trajectories from user prompts. The idea is to pass on the prompt to an LLM, find a starting anchor location (searched and refined using initial images used to construct the radiance field) and then use CineGPT (a cross-model transformer) trained for the next token prediction on quantized camera trajectories. Compared with some minimal baselines, the proposed approach attains favourable results in terms of MSE errors (on translation and rotation) and user-reviewed visual quality and alignment metrics.

**Strengths:**

- The paper is well written and well presented. I was able to understand the parts "explained in the paper" quickly. The figures are well made, and the videos in the supplementary section were easy to consume.

- The idea is well-motivated and I felt the work is in a good direction.

- I liked the idea of next token prediction on quantized camera trajectories and it appears novel to me.

**Weaknesses:**

- Baselines could be better; I feel the comparisons were extremely weak

- Many details in the paper are missing, making it difficult to comprehend the approach fully. The major misses are the details on the dataset construction, the proper explanation of baselines, and the LLM prompts. I give more detailed questions on the same in the next section.

- related work can be strengthened as well. They can cut a bit on the literature of radiance fields and 3D scene understanding and probably  add more on trajectory learning/optimization

**Questions:**

- It appears a path prediction network might do well in this case. What is eventually needed is to select a set of anchor locations on the path and then interpolate between them. It is not clear which interpolation algorithm was used for the baselines in the paper, which I believe is a crucial detail. Also, please bring more insights on which aspect of the baselines actually fails, is it the keypoint/keylocation selection or the interpolation.

- I would argue for replacing CineGPT with a rule-based optimizer or an interpolation algorithm. Since it does not observe visual modality, why is it not possible to do it through rule-based system (quantizing key kinds of camera movements).

- It is not clear to me, how to select the extent of camera movement. For example, in Figure 2, after determining the initial anchor (outside the window), CineGPT is called to pan straight forward. What aspect of the algorithm decides how long this pan should go? What stops it from bumping into walls, if the visual modality is not seen or explicit start-end points are not given?

- It is not clear how the dataset was constructed. Was it done in scenes with objects/people present in the CG? Who collected the dataset, and what instructions were given to the collector? How the collection was planned? What kind of camera movements were included? All these details are extremely crucial.

- There are open-source blender movies (e.g. https://spring-benchmark.org/) which are utilized in several benchmarks. Would it be a good idea to exploit these in the dataset construction?

- I would argue for an ordered keypoint prediction network and then spline interpolation between them. It would be a well designed meaningful baseline in this case.

- Not sure if clip can efficiently perform the task of location selection, would like to hear more from the users on the same

- Not fully clear, why would the method generalize to out-of-domain objects/places (for example, an opera house in this case. was it included in the train set?). That reduces the confidence on the final presented output.

- It would be useful to share the used prompt into the author's response, which would help the reviewers to try it out themselves and observe its limitations

- Discussion on some works on camera trajectories would be useful. Professional camera paths are often composed of constant, linear and parabolic segments [A][B]? The static trajectories also play a key role in this? Did the authors think about the importance of static trajectories and did they include it in the dataset?

- The final trajectory composition step appears to be non trivial to me and not fully clear at this stage. It looks like a hack right now.


[A] Grundmann et al. Auto-Directed Video Stabilization with Robust L1 Optimal Camera Paths. CVPR 2011
[B] Gandhi et al. Multi-Clip Video Editing from a Single Viewpoint. CVMP 2014

**Limitations:**

The authors did not discuss the limitations properly. I do not see an explicit limitations section.

---

> ### Author Rebuttal · Authors · 2024-08-07
>
> Thank you for your valuable feedback and insights. We appreciate your comments and would like to address your concerns in the following responses.
>
> **Baselines & Comparisons**
> - **Baseline Interpolation Algorithm.** We use cubic spline interpolation for translations and spherical linear interpolation (SLERP) for rotations, fixing camera intrinsics since baselines don’t account for them.
>
> - **Baseline Failures.** Baselines fail due to their limited understanding of text prompts, missing concepts like orientation ("turn left") or shape ("S-shaped"). Simple keypoint interpolation often produces incorrect results.
>
> - **Path Prediction Network.** We surveyed path prediction networks but found no related work. Specific references from the reviewer would be appreciated for comparison.
>
> - **Ordered Keypoint Prediction Network + Spline Interpolation** closely aligns with the approach used by ChatCam and our selected baselines, predicting keypoints through the Anchor Determinator or 3D latent embeddings, followed by cubic spline interpolation.
>
> - **Rule-based System** We built a rule-based system based on the suggestion. generating sub-trajectories from text via LLM, using set rules (1-to-1 mappings from text to trajectory) instead of ChatGPT. It predicts key locations using LERF.  Quantitative results are in the table (as suggested by reviewer **dvT5**, reporting metrics computed against multiple GTs for better fairness). **Qualitative results are in Figure C of the attached PDF.** The rule-based system performs worse than our baselines, as it cannot handle complex text descriptions like “go through the tunnel.”
> | Method          | Translation MDGT (↓)  | Rotation MDGT (↓)  |
> |-----------------|-----------------------|--------------------|
> | Rule-based System | 23.2                | 5.5                |
> | Ours             | 4.7                 | 2.1                |
>
> - **Weak Comparisons.** We argue that “as the first method to enable human language-guided camera operation, there is no established direct baseline for comparison.” To our knowledge:
>     1) No method achieves text-to-trajectory translation like CineGPT.
>     2) No multimodal-LLM system allows user-guided trajectory generation and optimization through conversation.
>
>     Therefore, we respectfully point out that stating “the comparisons were extremely weak” is somewhat unfair, as such interactions are not achievable by any baselines.
>
>
> **Extent of Camera Movement.** To keep the pipeline simple, we do not process scale specially. The trajectory scale largely depends on CineGPT's training data when the user doesn’t provide end anchor clues. This explains pan length decisions. Without clear visual modality or start-end points, bumping into walls is possible. Users can guide ChatCam to adjust trajectory scale by providing more anchor points through dialogue, as shown in the **conversation example provided in our general response**.
>
> **Dataset.** Our text-trajectory dataset contains trajectories and text descriptions, not attached to specific objects or scenes. Anchor determination bridges trajectories and specific scenes by placing trajectories in scenes through anchor points.
>
> - **“Why would the method generalize to out-of-domain objects/places?”** CineGPT and its training data are not tied to objects or places (no "domain" concept). Scenes like the opera house are recognized by the Anchor Determinator and its underlying CLIP model, allowing our method to work on them.
>
> - **Collection Details.** Data was collected by the authors. We enumerated textual descriptions of camera trajectories and built them in Blender. Movements included basic translations, rotations, focal length changes, and combinations (simultaneous or sequential). We included trajectories from professional cinematography literature. All the trajectories cover similar volumes with aligned translations and rotations.
>
> - **Blender Movies.** Since our dataset is not attached to any objects or places, suggested movies cannot be directly used in the dataset construction. However, they might be a meaningful application scenario for our future study with dynamic scenes.
>
> **CLIP-based Anchor Determination.** We use CLIP to select the image that best matches the text description of the anchor point, as it effectively aligns visual and textual information. CLIP can be replaced by other multi-modal models like BLIP. Our anchor refinement ensures key location accuracy. We verify CLIP-based anchor determination accuracy in the table below. We manually specified the optimal anchor points as ground truth in 10 cases and calculated the MSE between the Anchor Determination results and the ground truth.
>
> | Approach              | Anchor Error (↓) |
> |-----------------------|-------------------|
> | CLIP w/ refinement    | 3.7            |
> | BLIP w/ refinement    | 3.9          |
> | CLIP w/o refinement   | 8.5             |
>
> These results show that the choice of multi-modal model has minimal impact on accuracy, with both models performing well. However, anchor refinement greatly enhances anchor determination accuracy.
>
> **LLM Prompt.** We post our designed prompt in the general official comment, containing detailed instructions, response template, and examples for LLM. We encourage reviewers to test it with LLMs to better understand our approach. We will also make it public upon acceptance.
>
> **Static Trajectories.** Our dataset does not contain static trajectories, and thus CineGPT fails to produce correct results when prompted with "hold still" or something close. This issue can be addressed by adding static trajectories to the dataset. We will discuss works on camera trajectories in the related works as suggested.
>
> **Trajectory Composition.** We elaborate on this step in the official comment below.
>
> **Limitations.** We have discussed the limitations of our method in the appendix and will add a discussion of supporting static trajectories as suggested.

---

> ### Author Response · Authors · 2024-08-07
> **Trajectory Composition**
>
> The algorithm for trajectory composition works as follows:
> 1. It takes as input a list of trajectories and anchor points. If adjacent anchor points are encountered in the input, the algorithm will report this as illegal input.
> 2. Merge adjacent trajectories until there are no adjacent trajectories left in the list. During the merging process, calculate a Euclidean transformation (6 DoF) and apply it to the latter trajectory, ensuring its starting point coincides with the position and rotation of the endpoint of the previous trajectory.
> 3. For all trajectories, if a trajectory has two adjacent anchor points, find a similarity transformation (7 DoF) to make the position of its starting point and endpoint coincide with the two anchor points. If it has one adjacent anchor point, calculate a Euclidean transformation (6 DoF) to make the position and rotation of its starting or ending point coincide with this anchor point.
>
> We will include pseudo-code for this algorithm in future versions to enhance clarity.

---

> ### Comment · Reviewer_kHgE · 2024-08-14
> **On rule based trajectories**
>
> - As you created the trajectories in the dataset, in similar way you could create a dictionary of camera movements and give them a name (S-shaped, U-turn). Add few parameters to control and then use them across anchor points. Seems doable to me. Optionally, apply some optimization to post-process (example: L1 norm optimization or smoothing)
>
> - There are too many stochastic components and the reliability of the final result is doubtful
>
> I am still not convinced and I am staying with my original rating.

---

> ### Author Response · Authors · 2024-08-14
> **Thanks for your response**
>
> Thank you for your feedback!
>
> Your suggestion of "creating a dictionary of camera movements and giving them names (S-shaped, U-turn)" aligns with our understanding and implementation of the rule-based baseline during the rebuttal. However, as shown in the attached PDF and discussed with Reviewer **Shd5**, the results from this approach do not perform as well as our proposed method.
>
> This rule-based approach relies on a predefined dictionary of camera movements, which is inherently **finite**, whereas our GPT-based text-to-trajectory translation model can handle theoretically infinite input prompts.
>
> Additionally, we want to remind the reviewer that the rule-based approach cannot determine **anchor points** independently. The accurate determination of anchor points is a key part of our technical contribution. Moreover, the rule-based method lacks the ability to interact with users to modify or improve results, which is another significant advantage of our proposed approach.
>
> Regarding your concerns about the reliability of our final results, we respectfully disagree with the assertion that there are "too many stochastic components" in our approach. We welcome the reviewers to point out specific stochastic elements in our method so we can engage in further discussion.
>
> Before receiving your feedback shortly before the discussion period ends, we had already provided the LLM prompt as per your suggestion, allowing the conversational process in our method to be fully reproducible. **We encourage reviewers to try it out.** Additionally, any aspects of the final results that the user finds unsatisfactory can be adjusted through further interaction. Therefore, we believe that our results are reliable, especially when compared to rule-based approaches.

---

### Author Rebuttal · Authors · 2024-08-07

We want to thank all the reviewers for their time and insightful feedback regarding our first attempt at empowering LLMs beyond 1D NLP to understand 3D spatial relationships, specifically operating 3D camera trajectories in this paper. We are encouraged by the positive reception of our motivation, recognized as a good direction (**kHgE**), useful with high potential impact (**dvT5**), and making good use of LLMs (**Shd5**). Our submission was considered easy to follow, with well-made figures and videos (**kHgE**, **Shd5**, **oNWk**). Our proposed method was deemed novel (**kHgE**), creative (**dvT5**), clever (**Shd5**), and technically solid (**oNWk**). The results were regarded as excellent, demonstrating a wide range of trajectories on complex scenes (**dvT5**), and proving the efficiency of this method (**Shd5**). On the other hand, we acknowledge and will first address common concerns, followed by detailed responses to individual reviewers.

**Our Purpose.** Our multimodal-LLM approach allows natural language "chat" to instruct the 3D "camera" on reasoning in a complex 3D world. Compared with conventional tools that operate with 3D rotation, translation, camera intrinsics, keypoints, etc., ChatCam enables lay users to produce production-quality cinematography without requiring such technical knowledge. Users can engage in further conversation with ChatCam to interactively and iteratively refine the target camera trajectory, much like how a human director gives natural language suggestions to a human cinematographer in a movie post-production process.

Moreover, beyond its camera applications, our multimodal-LLM approach showcases the potential of conversational AI in multimodal reasoning and planning. This capability is crucial for developing embodied AI agents. We believe our work will not only be influential in its direct applications but also significantly advance the frontiers of the field.

**LLM Prompt.** As suggested by reviewer **kHgE**, we post in the official comment below our designed prompt to instruct the LLM agent. In this prompt, we provide the LLM with detailed instructions and guidelines for tool usage to achieve the target. We also include a template and examples for the LLM's responses. We encourage reviewers to test it out with LLMs on their own to better understand our approach. We will also make it public upon acceptance.

**Complex Case Example with User Interaction.** We provide a new example to verify the advantages of our method. As suggested by reviewer **dvT5**, we post the complete conversation between the user and ChatCam in the comment below. We include the visualization of trajectories and rendered video frames in Figures A & B of the attached PDF.
In this example, with a complex large-scale indoor scene, we asked ChatCam to generate a trajectory according to a detailed requirement. Initially, ChatCam's result goes directly through the wall. We then guide ChatCam to use new anchor points to avoid collision (concerned by **Shd5**) and successfully correct the trajectory. Finally, we extend this trajectory with a dolly zoom.

This trajectory involves panning downstairs, passing through a tunnel, performing a U-turn, and executing a dolly zoom within a complex room. We believe such intricate operations are not easily achievable by lay users through professional software and plug-ins with UI interfaces. This example demonstrates how our method allows user modification and interaction, and generates long and complex trajectories in intricate scenes, addressing concerns raised by reviewers **Shd5** and **oNWk**.

---

> ### Comment · Reviewer_Shd5 · 2024-08-12
> **Confusion about the Figure C.**
>
> In my opinion, you replacing CineGPT with a rule-based algorithm in the Figure C. So the anchor points should be same for these two methods. Figure C does not show the anchor 'entrance of the green tunnel'. CineGPT generate camera trajectories only based on text description without spatial information. How can you avoid collision when calling CineGPT with 'move forward through the tunnel'? I think if the anchor point is accurate, the collision will not occur with a high probability. In the same time, this will also be easy for rule-based method too.

---

> > ### Author Response · Authors · 2024-08-12
> > **Thanks for your response**
> >
> > Thank you for reading our rebuttal and for your question.
> >
> > In our implementation, we replaced the entire system (CineGPT + Anchor Determinator + LLM agent) with a rule-based algorithm, rather than just replacing CineGPT. Specifically, we used a 1-to-1 rule-based mapping to generate two sub-trajectories from the instructions "pan downwards" and "move forward." We then assembled these sub-trajectories and positioned them at the estimated first-frame key location ("top of the stairs"). This key location was estimated using LERF instead of our proposed Anchor Determinator. We made this choice as the rule-based system, without the LLM agent, lacks the capability to extract necessary inputs for anchor determination, such as "the entrance of the tunnel."
> >
> > While the "move forward" instruction can be executed correctly if the anchor is accurate, this assumption does not hold for the rule-based baseline. In fact, the rule-based approach cannot understand specific object-level prompts unless those objects are covered by the predefined rule set. In other words, this rule-based baseline can only generate a trajectory based on the input but lacks the ability to accurately place it within the scene. In this case, human assistance was required to determine the first key location, whereas our approach is capable of doing this autonomously.
> >
> > Additionally, we would like to note that different prompts were used for Figures A/B and Figure C. This was necessary because the input for Figures A/B would have led to meaningless results with the rule-based baseline. The rule-based baseline can only handle simple inputs like "pan forward," while our approach can manage much more complex inputs without human assistance to specify key locations. It can also collaborate with users to refine the results through conversation.
> >
> > We hope this clears up your confusion. We welcome you or other reviewers to raise any further concerns so that we can address them in the discussion period.

---

> > > ### Comment · Reviewer_Shd5 · 2024-08-13
> > >
> > > I appreciate the efficiency of your whole pipeline (user input -> LLM agent -> Anchor Determinator -> CineGPT). But while comparing the CineGPT with the rule-based/interpolation method, I think you should use the anchors generated from LLM agent and Anchor Determinator as the input of the rule-based and interpolation method, which can directly show the efficiency of the CineGPT.
> > >
> > > When the anchors are accurate, I am curious about the differences between CineGPT and the rule-based/interpolation method. For "move forward", I think it is easy to generate a straight trajectory based on rules and interpolation and adjust the scale to fit the start and end anchors.
> > >
> > > I am not sure if the specific object-level prompts are useful for CineGPT, according to "Since our dataset is not attached to any objects or places, suggested movies cannot be directly used in the dataset construction. ". Is there difference between the prompt "move forward through the tunnel" and "move forward along the road". I think the key of success is the spatial accuracy of the anchors.

---

> > > > ### Author Response · Authors · 2024-08-13
> > > > **Thanks for your response**
> > > >
> > > > Thank you for your detailed feedback!
> > > >
> > > > Based on your comments, we believe you agree that the entire ChatCam pipeline is efficient. This reinforces the significance of our study on conservation AI in camera control.
> > > >
> > > > We appreciate your suggestion to compare CineGPT with a rule-based approach while keeping the anchors the same, as this directly demonstrates the effectiveness of CineGPT. In the revised version of our paper, we will include both qualitative and quantitative results from this experiment.
> > > >
> > > > Since we are unable to modify the PDF at this stage, we will verbally describe the results of the rule-based/interpolation method in this scenario: with accurate anchor points, the rule-based approach successfully goes through the tunnel with the "move forward" prompt. However, it is also important to emphasize that the accurate determination of anchor points is not trivial and is a significant part of our contribution.
> > > >
> > > > Furthermore, the performance of the rule-based approach depends on the rule set, while the number of possible rules is theoretically infinite. If the user input includes more complex instructions beyond the predefined rule set, the rule-based method cannot perform effectively.
> > > >
> > > > We hope this can address your remaining concern regarding whether CineGPT could be replaced by a rule-based method. We would be happy to engage in further discussion on this or any other aspect.

---

> > > > > ### Comment · Reviewer_Shd5 · 2024-08-13
> > > > >
> > > > > Thanks for your response. My concerns have been addressed. It's good to add the ablation of the CineGPT and rule-based/interpolation method in the final version to show the key design of your pipeline.

---

### Author Response · Authors · 2024-08-07
**LLM Prompt （1/2）**

As suggested by reviewer **kHgE**, here we post our designed prompt to instruct the LLM agent.

You are a dialog agent that helps users to operate cameras in 3D scenes using dialog. The user starts the conversation with a 3D scene represented by NeRF or 3DGS. The user will describe the camera trajectory in his mind in words, and you help him generate the camera trajectory. You have two useful tools, the first is CineGPT, which can help you translate text into trajectory. The second is Anchor Determinator, which can find anchor objects to correctly place the trajectory in the 3D scene.

Please act according to the following instructions:
INSTRUCTIONS:
1.	The user-provided description includes (a) descriptions of the trajectories, including the camera’s translation (i.e., “pan forward”), rotation (i.e., “turn left”), and camera parameters (i.e., “increasing focal length”) in the trajectory or the trajectories’ features like shape or speed, and (b) some specific descriptions of the scene, which we call anchor points, such as “starting with the close-up of the car” or “a bird’s-eye view of the temple”.
2.	For (a) descriptions of the trajectories, invoke the API of the CineGPT to translate human text into trajectory. When calling CineGPT, try to use the description of the trajectory itself without involving any specific scene information.
i.	To summon CineGPT, the command is termed "infer_cinegpt". Its arguments are: "traj_description": "<traj_description>".
ii.	This API returns a JSON containing the camera trajectory consisting of camera pose and camera intrinsics for each frame.
3.	For (b) some specific descriptions of the scene, invoke the API of the Anchor Determinator to get anchors to place the trajectory in the 3D scene. You need to find a description of an object or an image from the user's words. Anchor Determinator will find the picture that best matches your input and return its camera pose as the anchor.
i.	To call Anchor Determinator, the command is termed "get_anchor". Its arguments are: "anchor_description": "<anchor_description>".
ii.	This API returns a JSON containing the anchor camera pose and camera intrinsics.
4.	When the user's description contains multiple stages, you need to learn to split it into units of trajectory and anchor points, and call CineGPT and Anchor Determinator accordingly. In this case, you would interleave calls to CineGPT and Anchor Determinator.
5.	Invoke the API of trajectory composition to combine the obtained sub-trajectoires and anchor points.
i.	To call trajectory composition, the command is termed "traj_compose". Its arguments are: " compose ": "<list_of_traj_anchor>".
ii.	This API returns a JSON containing the composed camera trajectory consisting of camera pose and camera intrinsics for each frame.
iii.	When encountering illegal input, this API will raise an error.
6.	Your generated plan should follow these steps.
i. Call CineGPT with trajectory descriptions (one or more times).
ii. Call Anchor Determinator with anchor descriptions (one or more times).
iii. Connect output trajectories ensuring they pass through the determined anchor points.
iv. Ask the user for further clarification if needed.
v. Render a video using the generated camera trajectory.

---

### Author Response · Authors · 2024-08-07
**LLM Prompt (2/2)**

You should only respond in JSON format following the template below:

RESPONSE TEMPLATE:
{
"observation": "observation",
"reasoning": "reasoning",
"plan": "a numbered list of steps to take that conveys the long-term plan"
}

In your response JSON, it is generated in the order of "observation", "reasoning", and "plan". Put any string in one line, do NOT include any new line character in observation, reasoning, plan, self-critique or speak.

Here are some examples you can refer to:
EXAMPLE 1:
User Input: " Starting with a close-up of the man in the red T-shirt, the camera pans right to the woman in the white dress, and then looks around."
Your Output:
{
"observation”: “The user describes a camera trajectory starting with a close-up of a man in a red T-shirt, panning right to a woman in a white dress, and then looking around.",
"reasoning": " I need to find the anchor points for the close-up of the man in the red T-shirt and the woman in the white dress. Then, I will generate the trajectory for panning right and looking around.",
"plan": " 1. Call Anchor Determinator with 'close-up of the man in the red T-shirt'. 2. Call Anchor Determinator with 'the woman in the white dress'. 3. Call CineGPT with 'pan right'. 4. Call CineGPT with 'look around'. 5. Connect the output trajectories, ensuring they pass through the anchor points. 6. Ask the user for further clarification if needed. 7. Render a video using the generated camera trajectory."
}

EXAMPLE 2:
User Input: "Take a dolly zoom shot of the woman's face."
Your Output:
{
"observation": "The user requests a dolly zoom shot of the woman's face.",
"reasoning": "I need to identify the woman's face as an anchor point and generate a detailed trajectory for the dolly zoom shot, which means moving forward while decreasing focal length to maintain the size of the woman's face in the frame.",
"plan": "1. Call Anchor Determinator with 'the woman's face'. 2. Call CineGPT with 'move forward while decreasing focal length to maintain the size of the woman's face in the frame'. 3. Connect the output trajectories, ensuring they pass through the anchor points. 4. Ask the user for further clarification if needed. 5. Render a video using the generated camera trajectory."
}
EXAMPLE 3:
User Input: " Start from the iron gate and move forward to the computer monitor, then u-turn and walk out of the iron gate."
Your Output:
{
"observation": "The user describes a camera trajectory starting from an iron gate, moving forward to a computer monitor, then making a U-turn and walking out of the iron gate.",
"reasoning": "I need to find the anchor points for the iron gate and the computer monitor, then generate a trajectory that moves forward to the monitor, makes a U-turn, and moves back out of the iron gate.",
"plan": "1. Call Anchor Determinator with 'the iron gate'. 2. Call Anchor Determinator with 'the computer monitor'. 3. Call CineGPT with 'move forward'. 4. Call CineGPT with 'U-turn and move backward'. 5. Connect the output trajectories, ensuring they pass through the anchor points. 6. Ask the user for further clarification if needed. 7. Render a video using the generated camera trajectory."
}

---

### Author Response · Authors · 2024-08-07
**Conversation Example**

As suggested by reviewer **dvT5**, here we post the complete conversation between the user and ChatCam for the new result. Please refer to Figures A & B of the attached PDF for the visualization of trajectories and rendered video frames.

**User:** From the top of the stairs, pan downwards to the entrance of the green tunnel. Move forward to go through the tunnel. Make a u-turn and move forward to the yellow bulldozer on the table.

**ChatCam:** { "observation": "The user describes a camera trajectory starting from the top of the stairs, panning downwards to the entrance of a green tunnel, moving forward through the tunnel, making a U-turn, and moving forward to a yellow bulldozer on a table.", "reasoning": "I need to find anchor points for the top of the stairs, the entrance of the green tunnel, and the yellow bulldozer. Then, I'll generate trajectories for panning downwards, moving forward through the tunnel, making a U-turn, and moving forward to the bulldozer.", "plan": "1. Call Anchor Determinator with 'top of the stairs'. 2. Call Anchor Determinator with 'entrance of the green tunnel'. 3. Call Anchor Determinator with 'yellow bulldozer on the table'. 4. Call CineGPT with 'pan downwards'. 5. Call CineGPT with 'move forward through the tunnel'. 6. Call CineGPT with 'U-turn'. 7. Call CineGPT with 'move forward to the bulldozer'. 8. Connect the output trajectories, ensuring they pass through the anchor points. 9. Ask the user for further clarification if needed. 10. Render a video using the generated camera trajectory." }

**User:** This trajectory hits a wall! Starts all over again. But after you make a u-turn, try going to a colorful bulldozer, then to a white table with books on it, and finally to a yellow bulldozer.

**ChatCam:** {"observation": "The user indicates that the initial trajectory hits a wall and needs revision. The revised trajectory starts from the top of the stairs, pans downwards to the entrance of the green tunnel, moves forward through the tunnel, makes a U-turn, then moves to a colorful bulldozer, proceeds to a white table with books, and finally ends at a yellow bulldozer.","reasoning": "I need to find anchor points for the colorful bulldozer and the white table with books, in addition to the previously identified points. I'll regenerate the trajectory to include these new elements and ensure the path avoids any obstacles like walls.","plan": "1. Call Anchor Determinator with 'top of the stairs'. 2. Call Anchor Determinator with 'entrance of the green tunnel'. 3. Call Anchor Determinator with 'colorful bulldozer'. 4. Call Anchor Determinator with 'white table with books'. 5. Call Anchor Determinator with 'yellow bulldozer on the table'. 6. Call CineGPT with 'pan downwards to the entrance of the tunnel'. 7. Call CineGPT with 'move forward through the tunnel'. 8. Call CineGPT with 'U-turn'. 9. Call CineGPT with 'move to the colorful bulldozer'. 10. Call CineGPT with 'move to the white table with books'. 11. Call CineGPT with 'move to the yellow bulldozer'. 12. Connect the output trajectories, ensuring they pass through the anchor points and do not hit any walls. 13. Ask the user for further clarification if needed. 14. Render a video using the generated camera trajectory."}

**User:** Very good! After this, I hope to add a dolly zoom for the yellow bulldozer. Give me the final plan.

**ChatCam:** { "observation": "The user wishes to add a dolly zoom effect targeting the yellow bulldozer at the end of the revised camera trajectory.", "reasoning": "To incorporate the dolly zoom effect, I'll need to adjust the camera movement and focal length as it approaches the yellow bulldozer, ensuring that the size of the bulldozer remains consistent in the frame while creating a dramatic spatial effect.", "plan": "1. Call Anchor Determinator with 'top of the stairs'. 2. Call Anchor Determinator with 'entrance of the green tunnel'. 3. Call Anchor Determinator with 'colorful bulldozer'. 4. Call Anchor Determinator with 'white table with books'. 5. Call Anchor Determinator with 'yellow bulldozer on the table'. 6. Call CineGPT with 'pan downwards to the entrance of the tunnel'. 7. Call CineGPT with 'move forward through the tunnel'. 8. Call CineGPT with 'U-turn'. 9. Call CineGPT with 'move to the colorful bulldozer'. 10. Call CineGPT with 'move to the white table with books'. 11. Call CineGPT with 'move to the yellow bulldozer'. 12. Call CineGPT with 'dolly zoom on the yellow bulldozer'. 13. Connect the output trajectories, ensuring they pass through the anchor points, do not hit any walls, and incorporate the dolly zoom effect. 14. Ask the user for further clarification if needed. 15. Render a video using the generated camera trajectory." }

---

### Comment · Area_Chair_5Gzu · 2024-08-12
**Facilitating the author-reviewer discussion**

Dear Reviewer of the submission 684,

Thank you for your time and effort so far in reviewing this paper.

The author submitted the rebuttal to your review, and the author-review discussion period will end tomorrow (13th August). It would be great if you could check if the rebuttal addresses your concern or not. Essentially, we have mixed reviews on this paper and need your input. Thank you in advance.

Best regards,
Your AC of submission 684

---

### Decision · Program_Chairs · 2024-09-25

**Decision:**

Accept (poster)

**Comment:**

This paper proposes a novel method for camera trajectory via the language model. Reviewers have questions and doubts mainly about the experiments including the baseline and data being used.  The author did a good job of addressing the concerns of the reviewer with detailed explanations and clarifications. Although one reviewer is still concerned about the experiment setting, the overall rating from the rest of the three reviewers is positive. Therefore, I suggest to accept this paper as a poster.